# Interventions to reduce sedentary behaviour in adults with type 2 diabetes: A systematic review and meta-analysis

Siobhan Smith[1,2]*, Babac Salmani[1], Jordan LeSarge[2], Kirsten Dillon-Rossiter[1], Anisa Morava[1], Harry Prapavessis[1]

1 Department of Kinesiology, Faculty of Health Sciences, Western University, London, Ontario, Canada,
2 Schulich School of Medicine and Dentistry, Western University, London, Ontario, Canada

* ssmith2025@meds.uwo.ca

## Abstract

Treatment and management of Type 2 Diabetes (T2D) includes physical activity, nutrition, and pharmacological management. Recently, the importance of reducing and breaking up sedentary behaviour has become recognized. This review aimed to summarize and synthesize the effectiveness of interventions in reducing and/or breaking up sedentary behaviour and cardiometabolic biomarkers in adults with T2D. A study protocol was preregistered on PROSPERO (CRD42022357281) and a database search (PubMed, EMBASE, Scopus, Web of Science, PsycINFO, SPORTDiscus, CINAHL, and Cochrane Library) was conducted on 16/09/2022 and updated on 03/01/2024. This review followed PRISMA guidelines and study quality was assessed with the Cochrane risk of Bias Tools. Twenty-eight articles were included in the review. The meta-analysis of short-term (Range: 3 hours– 4 days) sedentary behaviour interventions found significant improvement in continuous interstitial glucose measured for 24 hours after the sedentary behaviour intervention compared to control (SMD:-0.819,95%CI:-1.255,-0.383,p<0.001). Similarly, there was a significant improvement in postprandial interstitial glucose after the sedentary behaviour intervention compared to control (SMD:-0.347,95%CI:-0.584,-0.110,p = 0.004). Ten out of eleven longer-term (Range: 5 weeks– 3 years) sedentary behaviour interventions improved at least one measure of sedentary behaviour compared to control. Eight out of eight longer-term sedentary behaviour interventions improved at least one cardiometabolic biomarker compared to control. Reducing sedentary behaviour, independent of physical activity, can improve glycemic control in adults with T2D. Further, sedentary behaviour may be a feasible/ sustainable behaviour change.

## Introduction

Sedentary behaviour (SB) is defined as any waking behaviour with a low energy expenditure ($\leq$1.5 metabolic equivalents) while in a sitting, reclining, or lying posture [1]. For adults 18–64 years, the Canadian Society of Exercise Physiology (CSEP) recommends limiting sedentary time to 8 hours or less, which includes: no more than 3 hours of recreational screen time and breaking up long periods of sitting as often as possible [2]. However, Canadian adults are

**Funding:** The author(s) received no specific funding for this work.

**Competing interests:** The authors have declared that no competing interests exist.

sedentary for an estimated average of 9.5 hours of their waking day [3]. Of that time, Canadian adults accumulate an estimated average of 3.2 hours of recreational screen time [3].

Umbrella reviews and systematic reviews have concluded that SB is detrimentally associated with numerous health outcomes such as all-cause mortality, cardiovascular disease (CVD) mortality and incidence, type 2 diabetes (T2D) incidence, cognitive function, depression, function and disability, physical activity (PA) levels, and physical health related quality-of-life [4–6]. Further, they have found that reducing or breaking up SB may benefit body composition and markers of cardiometabolic risk [6]. However, the certainty of evidence was low because of inclusion of non-randomized intervention studies [6].

Following the 2020 umbrella review by Saunders and colleagues [6] there have been further systematic reviews that have investigated the effect of reducing and/or breaking up SB on cardiometabolic health in healthy adults. Reducing and/or breaking up SB may improve vascular function as measured by flow-mediated dilation [7], reduce fasting blood glucose [8] and reduce peripheral blood pressure [9]. Additionally, systematic reviews have investigated the effect of reducing and/or breaking up SB on cardiometabolic health in combined healthy and non-healthy adult populations. Reducing and/or breaking up SB may improve post-prandial glucose, insulin sensitivity, and triglycerides [10], endothelial function [11], and various cardiometabolic biomarkers [12, 13]. Further, systematic reviews have investigated the effect of reducing and/or breaking up SB on cardiometabolic health in combined various clinical populations (overweight/ obese, T2D, cardiovascular, neurological/ cognitive, and musculoskeletal diseases). Reducing and/or breaking up SB in various clinical conditions is possible and can reduce cardiometabolic risk [14]. Reducing and/or breaking up SB in adults that represent key stages in the pathogenesis of T2D (healthy, overweight, obese, impaired fasting glucose, impaired glucose tolerance, metabolic syndrome, and T2D) may improve vascular function [15]. Lastly, reducing and/or breaking up SB in adults with or at elevated risk of T2D (people with impaired glucose tolerance, are overweight, or are obese) may improve blood pressure [16]. However, to our knowledge, no systematic reviews have investigated the effect of reducing and/or breaking up SB in solely adults with T2D.

T2D is a unique cardiometabolic clinical condition and thus, should be examined independently. T2D occurs when the pancreas does not produce enough insulin or when the body does not effectively use the insulin that is produced. It is characterized by the presence of hyperglycemia, high blood sugar. In Canada, T2D accounts for 90% of diabetes cases [17]. There are many adverse health complications of diabetes such as, CVD, acute coronary syndromes, heart failure, dyslipidemia, hypertension (high blood pressure), chronic kidney disease, retinopathy (eye damage), neuropathy (nerve damage), lower extremity complications (foot ulcers and foot amputations), sexual dysfunction and hypogonadism in men, higher risk of infections, and poor mental health. The risk of these complications can be minimized by effective blood sugar control [17]. Treatment and management of T2D includes PA, nutrition, and pharmacological management [17]. However, recently the importance of reducing and breaking up SB has become recognized.

Thus, the overarching purpose of this systematic review is to summarize, synthesize the effectiveness of, and appraise the quality of short and long-term interventions that aim to reduce and/or break up SB in adults with T2D.

## Materials and methods

This systematic review followed PRISMA guidelines suggested for the reporting of systematic reviews [18]. The protocol for this systematic review was registered on PROSPERO (registration number CRD42022357281) prior to its commencement.

## Search strategy

A comprehensive systematic search was conducted to identify relevant studies on September 16, 2022 and updated on January 3, 2024. Databases searched included PubMED, EMBASE, Scopus, Web of Science, PsycINFO, SPORTDiscus, CINAHL, and Cochrane Library. The databases were searched using a combination of controlled vocabulary (MeSH) and title/abstract key words related to exposure (i.e., "sedentary behaviour", "sedentary time", "sedentary lifestyle", etc.) and population ("type 2 diabetes", "diabetes mellitus", "non-insulin dependent diabetes", etc.). The search was kept broad to avoid missing any relevant studies. The search was developed with the assistance of a librarian at Western University. The complete search can be found in the supplementary information (S1 File).

## Inclusion/ exclusion criteria

The inclusion and exclusion criteria for this systematic review will be outlined using the PICOT format: Population, Intervention, Comparator/ context, Outcomes, and Types of studies.

## Participants

This systematic review included persons ≥18 years of all ages (i.e., adults and older adults) with no restrictions on cultural, racial, or sex-based characteristics. All participants had to have been diagnosed with T2D.

## Intervention

This systematic review included studies where the interventions' primary or secondary aim was to reduce and/or break up SB. SB was defined as sitting or lying behaviours during waking time that expend low levels of energy ($\leq$ 1.5 metabolic equivalents) [1]. Studies were included if SB was measured objectively (i.e., with a device) or subjectively (i.e., self-reported measure). According to the definition of SB, SB can encompass a variety of activities including technology use, socialising, travel, and reading [1]. Thus, we included behaviour typical of sitting even if the study did not specify that it was a sitting activity (i.e., television viewing). Studies were excluded if they defined and/or measured SB as a lack of PA.

## Comparator/context

Studies from any geographic location or setting were included. Studies with any or no comparators were also eligible for inclusion.

## Outcomes

Studies including an outcome of objectively or subjectively measured total or domain specific SB (i.e., total daily sedentary time, screen time, etc.) were included. Studies including a cardiometabolic biomarker outcome (i.e., blood pressure, blood sugar, insulin sensitivity, lipids, etc.) were included. Studies that did not reported an outcome of SB and/or cardiometabolic biomarker were excluded.

## Type of studies

To be included in the present review, studies had to be interventions that aimed to reduce and/or break up SB in adults with T2D. Only interventional designs were included (i.e., randomized control trial, non-randomized control trial, crossover trials, etc.). Additionally,

observational studies (i.e., cross-sectional, longitudinal, retrospective, cohort, case-control, etc.), qualitative studies, reviews, commentaries, protocols, news articles and conference abstracts were excluded.

## Selection of studies

The articles retrieved by the search were imported into Covidence systematic review software. Titles and abstracts of the identified studies were independently checked by two review authors (SS and BS). All remaining eligible full texts were then screened independently by two review authors against inclusion criteria (SS and BS). At both screening stages, any discrepancies were settled by the third reviewer (JL). Reference lists of all included studies and relevant systematic reviews were screened by one reviewer (SS) for potential additional studies.

## Data extraction and synthesis

Data from each of the included reviews were extracted independently by one review author (SS) and checked by a second author (BS) for accuracy. The following data from each of the included studies were extracted: authors, year of publication, objective, participants, research design, intervention and comparison, SB outcomes, cardiometabolic outcomes, and results.

## Quality assessment

Two review authors (SS and BS) assessed the quality of each review independently using the Cochrane risk of Bias Tools [19]. Any discrepancies were settled by the third reviewer (JL). The RoB2 was used for randomized trials, which included using the version suitable for individually randomized parallel-group trials and the version suitable for crossover trials as appropriate. The RoB2 versions each contain 5 domains (1: risk of bias arising from the randomization process, 2: risk of bias due to deviations from the intended intervention, 3: risk of bias due to missing outcome data, 4: risk of bias in the measurement of the outcome, and 5: risk of bias in selection of the reported result) and an overall risk of bias judgement. Response options for individual questions include Yes/Probably Yes/Probably No/No/Not Indicated (Y/PY/PN/N/NI, respectively) and overall risk of bias judgement include Low/ High/ Some Concerns.

## Synthesis of results

Results of all short-term and longer-term interventions were summarized by narrative synthesis. Due to heterogeneity in objectives, participants, design, intervention/ comparison, SB, and cardiometabolic outcomes, all longer-term interventions in this review were only reported narratively, and no meta-analysis was conducted. Similar to Gardner et al., (2016), interventions were deemed 'promising' where there were either significant improvements in at least one measure of SB within the intervention group over time and/or improvements in at least one measure of SB in the intervention group compared to the control group [20]. Interventions were deemed 'non-promising' where there were neither significant improvements in SB within the intervention group over time nor significant improvements in SB in the intervention group compared to the control group.

Due to the homogeneity of all the short-term interventions in this review, they were also meta-analysed with a random-effects model using comprehensive meta-analysis software [21] by one reviewer (SS) and checked by a second reviewer (KDR). Glucose data via continuous glucose monitoring was extracted for the meta-analysis for both the 24-hours after the SB intervention and for any post-prandial periods assessed. Incremental area under the curve

(iAUC) for glucose was meta-analysed in preference to total area under the curve (tAUC), as iAUC is the recommended measure for detecting differences in post-prandial responses [22]. Mean and standard deviations or standard errors or 95% confidence intervals (CI) were extracted from individual studies and used to calculate the standardised mean differences (SMD). If the same data was reported in multiple publications, data was only extracted and used once. For the postprandial meta-analysis many studies reported multiple postprandial time periods (i.e., breakfast, lunch, and dinner), to account for all postprandial time periods, data from all the postprandial periods were combined for each intervention arm. Continuous outcomes were analysed using SMD to account for different measurement and time scales. When multiple sedentary break conditions were used in a study, data for all relevant conditions were synthesized and reported separately in the appropriate meta-analysis. When a study contained more than two intervention groups, (i.e., a control comparison was used twice in the same meta-analysis), the sample size for the control condition was divided by the number of times the control condition was used [21]. Pooled continuous data were expressed as SMD with 95% CI. SMDs were interpreted according to Cohen: 0.2 represents a small effect, 0.5 a moderate effect, and 0.8 a large effect [23].

Heterogeneity was assessed with the Chi-squared test ($p < 0.05$) $I^2$ statistic. Consistent with the Cochrane handbook for systematic reviews of interventions, the $I^2$ statistic was interpreted as 0–40%: might not be important; 30–60%: may represent moderate heterogeneity; 50–90%: may represent substantial heterogeneity; 75–100%: considerable heterogeneity [19]. Subgroup analysis was done for the 24-hour time period meta-analysis to examine any difference between 1-day interventions and multiday interventions. Meta-regression was done to assess the impact of moderators of age, sex, and BMI on the data. Publication bias was assessed visually by funnel plots.

## Results

A total of 13430 articles were retrieved from the databases searched (PubMed, Embase, Scopus, Web of Science, PsychInfo, SPORTDiscus, CINAHL, and the Cochrane Library) and nine articles from the reference lists of relevant systematic reviews, see Fig 1. After duplicates were removed, 7,238 articles remained. After screening the titles and abstracts, 168 articles were read in their entirety. Of those, 28 interventions met the inclusion and exclusion criteria and were included in the systematic review.

There were two main types of interventions identified. The first were short-term (Range: 3 hours—4 days) cross-over design interventions that manipulated SB over days to determine the effect on various cardiometabolic outcomes. The second type of interventions were longer-term (Range: 5 weeks– 3 years) that measured the effect of an intervention on SB with and/or without additional cardiometabolic outcomes.

### Short-term SB interventions in adults with T2D

There were 13 short-term SB interventions included in the systematic review consisting of seven unique studies. Sample sizes ranged from 12 to 30 participants, mean ages ranged from 60 to 65, and BMI ranged from 23.6 to 33.0. All studies except one included both female and male participants. All studies used a randomized crossover design. Most studies used 1-day conditions after the intervention, whereas three studies used 180min condition, 3-day, and 4-day conditions, respectively. The most common outcome assessed was blood glucose. The characteristics (objective, participants, research design, intervention/ comparison, cardiometabolic outcomes, and main results) of all eligible short-term interventions are found in supplementary information (S1 Table) and summarized in Table 1 below.

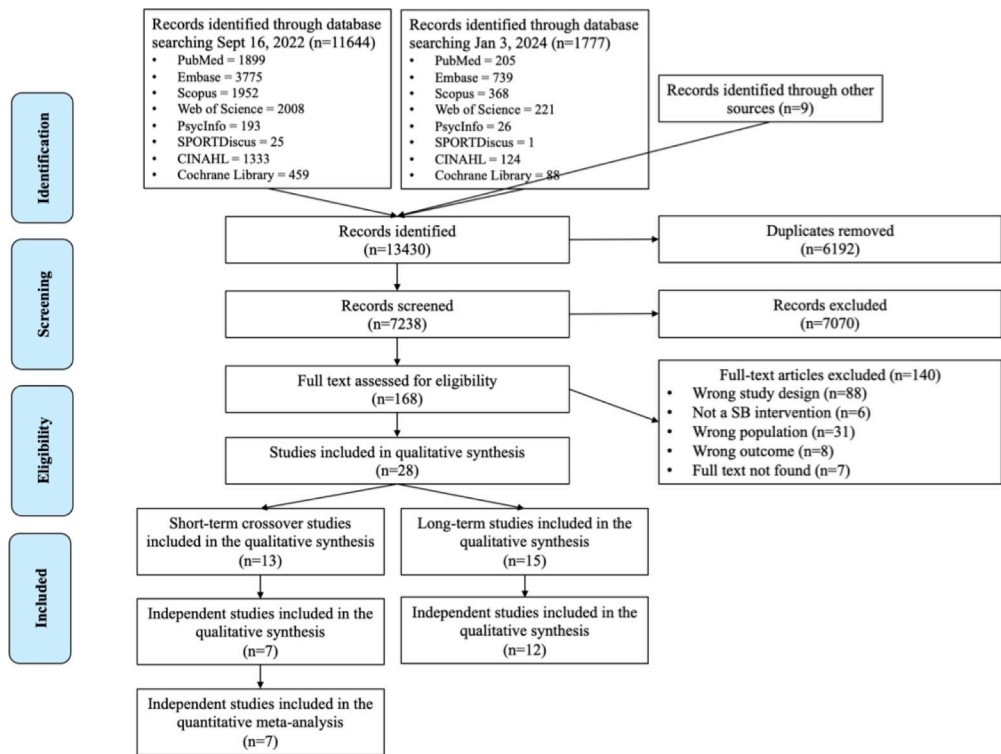

**Fig 1. PRISMA (Preferred reporting items for systematic reviews and meta-analyses) flow chart of the eligibility process.**

Non-glucose outcomes in the short-term interventions were not meta-analyzed and are described in Table 1. Of the short-term interventions that investigated insulin outcomes, 4 of 4 studies found significant improvements compared to control. Of the short-term interventions that investigated lipid outcomes, 2 of 3 studies found significant improvements compared to control. Of the short-term interventions that investigated vascular outcomes, 2 of 2 studies found significant improvements compared to control. Of the short-term interventions that investigated other outcomes, one study found significant improvements in mean plasma nor-adrenaline, and another found significant improvements in energy expenditure compared to control.

## Quantitative synthesis meta-analysis

In the random effects meta-analysis, shown below (see Fig 2), values left of 0 (i.e., negative values) represent improved 24-h continuous interstitial glucose for the intervention (breaking up sitting) condition over the control (sitting) condition. The SMD effect size is -0.819 with a 95% CI of -1.255 to -0.383. This effect is statistically significant p<0.001. According to Cohen (1988), $\geq |0.8|$ constitute a large effect.

In terms of heterogeneity, the Q-value is 28.993 with 9 degrees of freedom and p = 0.001. Thus, heterogeneity is statistically significant and true effect size is not the same in all studies. The $I^2$ statistic is 69%, which is considered substantial heterogeneity, 69% of the variance in observed effects reflects variance in true effects rather than sampling error [19]. Tau-squared, the variance of true effect sizes is 0.314 d units, Tau, the standard deviation of true effect sizes, is 0.561 d units. The prediction interval is -2.211 to 0.56, where the true effect size in 95% of all comparable populations falls in this interval.

**Table 1. Short-term SB interventions in adults with T2D.**

| Study | Participants | Intervention Design & Length | Glucose outcomes | Non-Glucose Outcomes | | | | RoB2 Quality |
|---|---|---|---|---|---|---|---|---|
| | | | | Insulin Outcomes | Lipid Outcomes | Vascular Outcomes | Other Outcomes | |
| **1** **(Blankenship et al., 2019)** [24] | N = 30 (n = 10/ condition 20, 40, or 60 min), mean age 64 ±8.2, 16 women [53.33%], mean BMI 31.7 | RCD = Three 24h experimental conditions: morning walk after breakfast (WALK), post-meal breaks from sitting (BR), and sedentary control (CON). | | | | | | Some Concerns |
| **2** **(Dempsey, Larsen, et al., 2016)** [25] **(Dempsey Sacre, et al., 2016)** [26] **(Dempsey et al., 2017)** [27] **(Grace et al., 2017)** [28] | N = 24, mean age 62 ±6, 10 women [41.67%], mean BMI 33.0 | RDC = Three 8h conditions: uninterrupted sitting (control), sitting plus 3-min bouts of LW every 30 min, and sitting plus 3-min bouts of SRA every 30 min. | | The LW and SRA conditions produced significant improved insulin and c-peptide compared with sitting control. | The LW and SRA conditions produced significant improved triglyceride (SRA only), diacylglycerols, triacylglycerols, phosphatidylethanolamines, plasmalogens, lysol alkyl phosphatidylserines (LW only) compared with sitting control. | The LW and SRA conditions produced significant improved SBP, DBP, and mean resting HR (LW only) compared with sitting control). | The LW and SRA conditions produced significant improved mean plasma noradrenaline compared with sitting control. | Some Concerns |
| **3** **(Duvier et al., 2017)** [27] | N = 19, mean age 63 ±9, 6 women [31.58%], mean BMI 30.5 | RCD = Three 4d activity regimens: sitting (14h of waking daily sitting), exercise (1h of sitting time replaced with MVPA), and sit less (4h sitting replaced with 2h LW and 3h standing, sitting broken up every 30 min). | | The sit-less condition produced significant improvements in insulin and c-peptide over the sitting condition. The sit-less condition produced significant improvements in insulin over the exercise condition. | The sit-less condition produced significant improvements in triacylglycerol, non-HDL-C, NEFA over the sitting condition. | | The sit-less condition produced significant improvements in energy expenditure over the sitting and exercise conditions. | High |

*(Continued)*

Table 1. (Continued)

| Study | Participants | Intervention Design & Length | Glucose outcomes | Non-Glucose Outcomes | | | | RoB2 Quality |
|---|---|---|---|---|---|---|---|---|
| | | | | Insulin Outcomes | Lipid Outcomes | Vascular Outcomes | Other Outcomes | |
| 4 (Homer, Taylor, Dempsey, Wheeler, Sethi, Townsend, et al., 2021) [29] (Homer, Taylor, Dempsey, Wheeler, Sethi, Grace, et al., 2021) [30] (Taylor et al., 2021) [31] | N = 24, mean age 62 ±8, 11 women [41.67%], mean BMI 32.7 | RCD = Three 8h conditions: uninterrupted sitting (control), sitting plus 3-min bouts of SRA every 30 min, sitting plus 6-min bouts of SRAR every 60 min. | | The SRA6 conditions produced significantly attenuated insulin compared with sitting control. No significant differences in insulin were observed in comparison of SRA3 and sitting control. | No significant differences of condition on triglyceride were observed. | SRA3 produced significantly higher FMD compared to SIT. SRA an dSRA6 produced significantly higher mean resting femoral shear rate compared to SIT. Endotthelin-1 concentrations were not statistically different between conditions. | | Some Concerns |
| 5 (Honda et al., 2016) [32] | N = 16, mean age 65 ±1.1, 3 women [18.75%], mean BMI 23.6 | RCD = Two 180min postprandial conditions: uninterrupted postprandial sitting (control) and interrupted postprandial sitting with 3 min bout of stair climbing descending exercise (ST-EX) at 60min and 120min. | | | | | | Some Concerns |

(Continued)

**Table 1.** (Continued)

| | Study | Participants | Intervention Design & Length | Glucose outcomes | | Non-Glucose Outcomes | | | | | RoB2 Quality |
|---|---|---|---|---|---|---|---|---|---|---|---|
| | | | | | | Insulin Outcomes | Lipid Outcomes | Vascular Outcomes | Other Outcomes | | |
| 6 | **(Paing et al., 2019a)** [33]<br><br>**(Paing et al., 2019b)** [34] | N = 12 (n = 8/ condition), mean age 60.0 ±3.2, 4 women [33.33%], mean BMI 30.2 | RCD = Three 7h sitting conditions: interrupted every 60 min (condition 1), interrupted every 30 min (condition 2), and interrupted every 15 min (condition 3) by 3-min bouts of LW. | | | | | | | | Some Concerns |
| 7 | **(Van Dijk et a., 2013)** [35] | N = 20, mean age 64 ±1, women n = 0 [0%], mean BMI 29.5 | RCD = Three 3d conditions: SB control, breaking up SB by three post-meal 15-min bouts of ADL (3 METs), and breaking up SB by a single 45-min bout of MVPA (6 METs) performed on day 2. | | | The ADL and exercise conditions produced significant improvements in insulin compared with the sedentary condition. | | | | | Some Concerns |

Note: Green indicates a significant effect of the intervention. Yellow indicates a non-significant effect of the intervention. Grey indicates that no outcome was assessed. Bold indicates the intervention focused on sedentary behaviour. RCD (randomized crossover design).

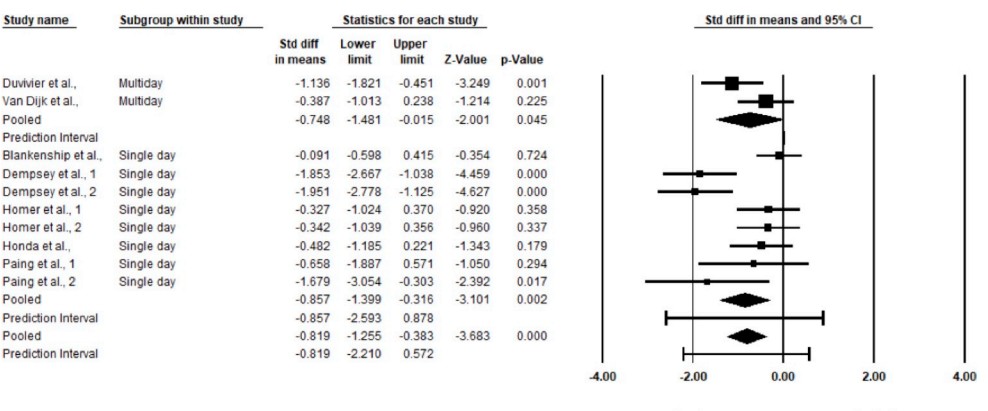

**Fig 2. Forest plot of the effects of intervention (breaks from sitting) vs control (sitting) on continuous interstitial glucose measured for 24-hours after the SB intervention.** Effect sizes are the SMD with a 95% CI.

In the meta-regression for age, sex, and BMI against the 24-hour glucose outcome, only age was significant p<0.05, where increased age showed increased effect of the intervention (breaking up sitting) on 24-hour continuous interstitial glucose (see Fig 3).

Evidence of publication bias is visible in Fig 4 given its asymmetrical nature. Note that heterogeneity, meta-regression, and publication bias based on less than ten studies is not likely to be reliable and thus, should be interpreted with caution.

In the random effects meta-analysis, shown below (Fig 5), values left of 0 (i.e., negative values) represent improved continuous interstitial postprandial glucose for the intervention (breaking up sitting) condition over the control (sitting) condition. The SMD effect size is -0.347 with a 95% CI of -0.584 to -0.110. This effect is statistically significant p = 0.004. According to Cohen (1988), $\geq |0.2–0.4|$ constitute a small effect [23].

In terms of heterogeneity, the Q-value is 7.136 with 8 degrees of freedom and p>0.05. Thus, heterogeneity is estimated to be 0 and is not statistically significant and true effect size is

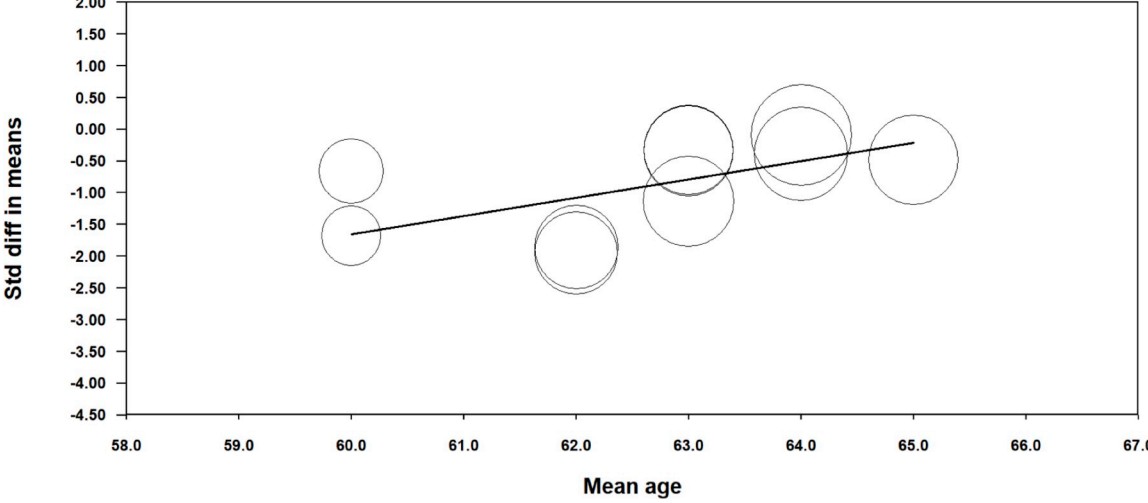

**Fig 3. Meta-regression for 24h continuous interstitial glucose and age.** Circles represent individual studies.

**Funnel Plot of Standard Error by Std diff in means**

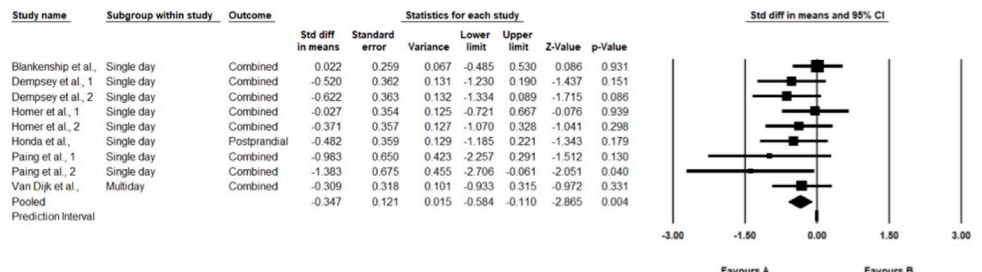

**Fig 4. Funnel plot for the 24h continuous interstitial glucose measures random-effects model.** Circles are individual effect sizes with the standardized mean difference on the x-axis and the standard error of the effect sizes on the y-axis. The vertical line represents the summary effect size estimate in the meta-analysis model.

estimated to be the same in all studies. The $I^2$ statistic is 0%, which is considered no heterogeneity, 0% of the variance in observed effects reflects variance in true effects rather than sampling error [19]. Tau-squared, the variance of true effect sizes is 0.000 d units, Tau, the standard deviation of true effect sizes, is 0.000 d units. Since tau-squared is estimated as 0, we assume that all studies share a common effect size, and there is no dispersion of true effects.

In the meta-regression age, sex, and BMI, were not significant $p > 0.05$.

Evidence of publication bias is visible in Fig 6 from its asymmetrical nature. Note that heterogeneity, meta-regression, and publication bias based on less than ten studies is not likely to be reliable and thus, should be interpreted with caution.

## Quality

The overall risk of bias for the short-term SB intervention in adults with T2D was *some concerns* (as per RoB2 scoring) for N = 12 and *high risk* (as per RoB2 scoring) of bias for N = 1. Notably, many studies failed to report if there were any deviations from the intended intervention, leading to the assigned risk of bias scores possibly overestimating as a higher risk of bias score. The overall RoB2 quality score can be found in Table 1. The complete quality assessment can be found in the supplementary information (S3 Table).

| Study name | Subgroup within study | Outcome | Std diff in means | Standard error | Variance | Lower limit | Upper limit | Z-Value | p-Value |
|---|---|---|---|---|---|---|---|---|---|
| Blankenship et al., | Single day | Combined | 0.022 | 0.259 | 0.067 | -0.485 | 0.530 | 0.086 | 0.931 |
| Dempsey et al., 1 | Single day | Combined | -0.520 | 0.362 | 0.131 | -1.230 | 0.190 | -1.437 | 0.151 |
| Dempsey et al., 2 | Single day | Combined | -0.622 | 0.363 | 0.132 | -1.334 | 0.089 | -1.715 | 0.086 |
| Homer et al., 1 | Single day | Combined | -0.027 | 0.354 | 0.125 | -0.721 | 0.667 | -0.076 | 0.939 |
| Homer et al., 2 | Single day | Combined | -0.371 | 0.357 | 0.127 | -1.070 | 0.328 | -1.041 | 0.298 |
| Honda et al., | Single day | Postprandial | -0.482 | 0.359 | 0.129 | -1.185 | 0.221 | -1.343 | 0.179 |
| Paing et al., 1 | Single day | Combined | -0.983 | 0.650 | 0.423 | -2.257 | 0.291 | -1.512 | 0.130 |
| Paing et al., 2 | Single day | Combined | -1.383 | 0.675 | 0.455 | -2.706 | -0.061 | -2.051 | 0.040 |
| Van Dijk et al., | Multiday | Combined | -0.309 | 0.318 | 0.101 | -0.933 | 0.315 | -0.972 | 0.331 |
| Pooled | | | -0.347 | 0.121 | 0.015 | -0.584 | -0.110 | -2.865 | 0.004 |
| Prediction Interval | | | | | | | | | |

**Fig 5. Forest plot of the effects of intervention (breaks from sitting) vs control (sitting) on continuous combined interstitial postprandial glucose measures.** Effect sizes are the SMD with a 95% CI.

Funnel Plot of Standard Error by Std diff in means

**Fig 6. Funnel plot for combined continuous interstitial postprandial glucose measures random-effects model.** Circles are individual effect sizes with the standardized mean difference on the x-axis and the standard error of the effect sizes on the y-axis. The vertical line represents the summary effect size estimate in the meta-analysis model.

## Longer-term SB interventions in adults with T2D

There were 15 longer-term SB interventions included in the systematic review consisting of 12 unique studies. Sample sizes ranged from 10 to 397 participants, mean ages ranged from 55 to 75, and BMI ranged from 24.7 to 33.6. All studies included both female and male participants. All studies, except two, used a randomized control trial design, one study used a non-randomized control design and the other a pre-post intervention design. Interventions ranged from five weeks to three years. Only three interventions focused solely on reducing and/or breaking up on SB. A variety of intervention methods were used. Sedentary outcomes were assessed subjectively by self-report (N = 3), objectively (N = 5), and both subjectively and objectively (N = 3). A variety of cardiometabolic outcomes were assessed (i.e., biochemical and anthropometric). The characteristics (objectives, participants, research design, intervention/ comparison, SB outcomes, cardiometabolic outcomes, and main results) of all eligible longer-term interventions can be found in supplementary information (S2 Table) and summarized in Table 2 below.

Due to the large heterogeneity between intervention design, duration, methods, outcomes, and other factors, a meta-analysis of longer-term SB interventions was not appropriate. Consequently, the results have been interpreted qualitatively. All interventions, except one, were deemed as 'promising'. For instance, for Alonso-Dominguez et al., (2019) [36], sedentary time significantly decreased for the intervention group at 3-months (p < .001) and 12-months (p < .001) and sedentary time was significantly different between the control and intervention group at 3-months (p < .05) but not at 12-months (p>.05). Brazo-Sayavera et al., (2021) [43] was the only intervention that did not show improvement in some measure of SB over time or when compared to control. They found that sitting time increased by 58% and 32% for control group (d = −1.84) and intervention group (d = −1.17), respectively (time × group interaction, P < .05). However, notably, sitting time increased more over time in the control group than the intervention group. Further, sitting time was self-reported, the intervention focused on PA rather than SB, the intervention was nonrandomized, and the intervention was rated as high risk of bias, so results should be interpreted with caution.

**Table 2. Longer-term SB intervention in adults with T2D.**

| | Study | Participants | Intervention Design & Length | Sedentary Outcome | Cardiometabolic Outcome | RoB2 Quality |
|---|---|---|---|---|---|---|
| 1 | (Alonso-Domínguez et al., 2019) [36] | N = 204 | 3-month RT with 1-year f/u | [green] | Glycemic/ Lipids/ Vascular/ Anthropometric | Some Concerns |
| 2 | (Althoman et al., 2021) [37] | N = 10 | 13-week pre-post intervention | O (ActivPAL) | Glycemic | High |
| 3 | (Balducci et al., 2017) [38] (Balducci et al., 2019) [39] (Balducci et al., 2022a) [40] (Balducci et al., 2022b) [41] | N = 300 | 3-year RCT | O | Glycemic/ Vascular/ Anthropometric/ Coronary Heart Disease & Stroke Risk | Some Concerns |
| 4 | **(Bailey et al., 2020)** **[42]** | N = 18 | 8-week RCT | O (ActivPAL) | Glycemic/ Anthropometric | High |
| 5 | (Brazo-Sayavera et al., 2021) [43] | N = 35 | 11-week non-RCT | [yellow] | [grey] | High |
| 6 | (Connelly et al., 2017) [44] | N = 31 | 6-month RCT | O | Glycemic | Some Concerns |
| 7 | (De Greef et al., 2010) [45] | N = 41 | 12-week RCT with 1-year f/u | O | Glycemic/ Vascular | Some Concerns |
| 8 | (De Greef et al., 2011) [46] | N = 92 | 24-week RCT with 1-year f/u | O | [grey] | High |
| 9 | **(Hsu et al., 2023)** **[47]** | N = 48 | 12-week RCT | [grey] | Glycemic/ Lipids | Some Concerns |
| 10 | (Jennings et al., 2013) [48] | N = 397 | 12-week RCT with 6-month f/u | [green] | [grey] | High |
| 11 | **(Miyamoto et al., 2017)** **[49]** | N = 31 | 12-week RCT | O (ActivPAL) | Glycemic | High |
| 12 | (Poppe et al., 2019) [50] | N = 54 | 5-week RCT | O | [grey] | High |

Note: Green indicates a promising effect of the intervention. Yellow indicates a non-promising effect of the intervention. Grey indicates that no outcome was assessed. Bold indicates the intervention focused on sedentary behaviour. O indicates that sedentary outcomes were assessed objectively. RCT (randomized controlled trial). f/u (follow-up). N = number of participants.

All interventions that assessed cardiometabolic markers were deemed as 'promising'. For instance, for Alonso-Dominguez et al., (2019), [36] postprandial glycaemia, lipid profile, and

systolic blood pressure significantly (p < .05) improved for the intervention group at 3-months. LDL cholesterol, anthropometric parameters, and systolic blood pressure significantly (p < .05) improved for the intervention group at 12-months. BMI and waist circumference were significantly different (p < .05) between the control and intervention groups at 3-months. No significant differences were found between the groups at 12 months (p>.05).

## Quality

The overall risk of bias for the longer-term SB interventions in adults with T2D was *some concerns* (as per RoB2 scoring) for N = 5 and *high risk of bias* (as per RoB2 scoring) for N = 7. Notably, many studies failed to report if there were any deviations from the intended intervention, leading to the assigned risk of bias scores possibly overestimating as a higher risk of bias score. The overall RoB2 quality score can be found in Table 2. The complete quality assessment can be found in the supplementary information (S3 Table).

## Discussion

Previous umbrella reviews and systematic reviews have concluded that SB is detrimentally associated with numerous health outcomes [4–6]. Further, they have found that reducing and/ or breaking up SB can improve numerous health outcomes [6, 7, 12, 14, 15]. However, to our knowledge, this is the first systematic review and meta-analysis to investigate the effect of reducing and/or breaking up SB in adults with the unique cardiometabolic condition of T2D.

The overarching aim of this systematic review was to summarize, synthesize the effectiveness of, and appraise the quality of short and long-term interventions that aimed to reduce and/or break up SB in adults with T2D. For the short-term SB intervention studies, we applied meta-analytic procedures to quantify the effect of the interventions on continuous 24-hour and postprandial interstitial glucose. We examined non-glucose outcomes through a narrative (qualitative) lens. For the long-term SB interventions, we again used a narrative approach to shed light on whether these interventions were effective in reducing and/or breaking up SB and improving cardiometabolic biomarkers (i.e., blood glucose, insulin sensitivity, etc.).

### Short-term SB interventions

A meta-analysis of the short-term SB interventions showed significant improvements in continuous interstitial glucose of the intervention (sit-less condition) over control (sitting). This is in agreement with Loh et al., (2020) [10] who conducted a systematic review and meta-analysis to compare the effects of breaking up prolonged sitting with bouts of PA (INT) versus continuous sitting (CON) on glucose in clinical and non-clinical populations. Researchers found significant SMDs in favor of the INT group for lower glucose measures. Further, Nieste et al., (2021) [14] investigated the effect of different lifestyle interventions on SB and cardiometabolic health in clinical populations (e.g. overweight/ obese, T2D, cardiovascular, neurological/ cognitive, and musculoskeletal diseases) and found that glycated hemoglobin concentration was significantly reduced overall in the intervention groups compared to control groups (- 0.17%; 95% CI: [- 0.30, - 0.04]%; *p* = 0.01).

However, the current meta-analysis consists of only seven unique short-term studies. Much longer randomized controlled trials aimed at reducing SB and measuring the effect on markers of cardiometabolic health, with a focus on blood glucose, in adults with T2D are needed to be confident that reducing SB can lead to meaningful improvements in glycemic control in adults with T2D. Currently, we recommend that adults with T2D aim to reduce their SB in order to improve their glycemic control.

Of the short-term non-glucose outcomes, 4 of 4 studies that investigated insulin outcomes, 2 of 3 studies that investigated lipid outcomes, 2 of 2 studies that investigated vascular outcomes, and 2 of 2 studies that investigated other cardiometabolic outcomes found significant improvements compared to control. These findings, taken together, offer evidence that short-term SB interventions can positively change a variety of non-glucose health outcomes in patients with T2D.

## Long-term SB interventions

Long-term SB interventions in adults with T2D to date appear to be able to reduce and/or break up sitting. Of the 12 interventions identified in this review, only one failed to significantly improve SB outcomes. This suggests that reducing and/or breaking up sedentary is likely an achievable behaviour change for adults with T2D. This is in agreement with a systematic review by Nieste et al., (2021) [14], which investigated the effect of different lifestyle interventions on SB in clinical populations. Nieste et al., (2021) [14] showed that participants with various clinical conditions (e.g. overweight/ obese, T2D, cardiovascular, neurological/ cognitive, and musculoskeletal diseases) reduced their overall SB by 64 min/day (95%CI: [–91, –38] min/day; $p < 0.001$), with larger within-group differences of multicomponent behavioural interventions including motivational counselling, self-monitoring, social facilitation, and technologies (-89 min/day; 95%CI: [–132, –46] min/day; $p < 0.001$).

Adults with T2D have always been a difficult population to accomplish behavioural change, including PA and nutrition. The results from the current systematic review are promising; SB changes may be more realistic, achievable, and sustainable goal for adults with T2D rather than PA or nutrition behaviour change. More research is needed to see if changes in sedentary behaviour can persist over a longer duration.

In the current systematic review, only three interventions focused solely on changing SB [42, 47, 49], rather than changing PA *and* SB. However, research highlights that interventions should focus on SB rather than PA or both PA and SB to optimizing changing SB [20, 51]. Further, two of three interventions that did focus on solely reducing SB were deemed to be at high risk of bias [42, 49]. Thus, more high quality randomized controlled trials in adults with T2D are needed that focus solely on changing SB in this population.

Further, three interventions used only self-reported measures of SB rather than solely objective or both objective and subjective measures [36, 43, 48]. Of those that did use an objective measure, only three reported using a tri-axial accelerometer capable of fully assessing SB, such as the activPAL [37, 42, 49]. The reason the activPAL is considered the gold standard for SB is due to its ability to distinguish between sitting and standing [52]. Future SB interventions should make sure to quantify SB changes objectively rather than just subjectively, and if possible, with a gold standard tri-axial accelerometer, such as the activPAL device. It also appears that interventions in adults with T2D that target SB to date improve cardiometabolic health, including better glycemic control. Of the longer-term interventions, 8 of 12 studies investigated at least one cardiometabolic outcome and all 8 found significant improvements in at least one cardiometabolic outcome overtime or when compared to control.

## Strengths and limitations

With respect to strengths, this is the first systematic review and meta-analysis to our knowledge that investigate the effect of reducing and/or breaking up SB in the cardiometabolic unique population of adults with T2D. Further, PRISMA guidelines were followed, a broad search was conducted with guidance from a university librarian, quality was assessed using the Cochrane Risk of Bias tools, and the review was pre-registered. Additionally, a meta-analysis

and meta-regression on the short-term effects of SB interventions on glucose control were conducted, and heterogeneity and publication bias were assessed. However, several limitations must be acknowledged. Selection bias is always a risk with systematic reviews. Only English articles were included in the review as the review authors are only fluent in English. Further, articles may have been missed in any of the search or screening stages. Also, many of the longer-term SB interventions focused on both PA or PA and SB. Future interventions should focus on SB rather than PA or both PA and SB, to optimize changing SB. Lastly, a meta-analysis could only be conducted for the short-term interventions and for only one cardiometabolic biomarker, glucose control, due to the paucity of research in this field and heterogeneity of the longer-term SB interventions. Once more randomized interventions are conducted, future meta-analyses need to quantify the optimal dose of SB reduction in a population of adults with T2D to improve other cardiometabolic biomarkers (i.e., glucose, insulin, vascular, anthropometric, etc.).

## Conclusions

This systematic review and meta-analysis found a statistically significant advantage of reducing SB in the short-term for improving glycemic control in adults with T2D. Further, it appears that longer-term SB interventions may both reduce SB and improve cardiometabolic health. However, more high-quality longer-term randomized controlled trials in adults with T2D are needed that focus solely on changing SB rather than both SB and PA and that quantify changes in SB objectively rather than subjectively. Further, future long-term SB interventions in adults with T2D should examine the effect of reducing SB on cardiometabolic health, particularly glucose control. In conclusion, at this time we recommend that adults with T2D aim to reduce and/ or break up their SB to improve their glycemic control.

## Supporting information

**S1 Table. Short-term SB interventions in adults with T2D.**
(DOCX)

**S2 Table. Longer-term SB interventions in adults with T2D.**
(DOCX)

**S3 Table. Quality of eligible interventions.**
(DOCX)

**S1 File. Database searches.**
(DOCX)

## Author Contributions

**Conceptualization:** Siobhan Smith, Harry Prapavessis.

**Data curation:** Siobhan Smith, Babac Salmani, Jordan LeSarge, Kirsten Dillon-Rossiter.

**Formal analysis:** Siobhan Smith, Kirsten Dillon-Rossiter.

**Investigation:** Siobhan Smith, Babac Salmani, Jordan LeSarge, Kirsten Dillon-Rossiter, Anisa Morava.

**Methodology:** Siobhan Smith, Babac Salmani, Jordan LeSarge, Kirsten Dillon-Rossiter, Anisa Morava, Harry Prapavessis.

**Project administration:** Harry Prapavessis.

**Resources:** Harry Prapavessis.

**Software:** Harry Prapavessis.

**Supervision:** Harry Prapavessis.

**Visualization:** Siobhan Smith.

**Writing – original draft:** Siobhan Smith.

**Writing – review & editing:** Siobhan Smith, Babac Salmani, Jordan LeSarge, Kirsten Dillon-Rossiter, Anisa Morava, Harry Prapavessis.

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
