## [Decision Letter · Decision Letter 0]

30 Apr 2024

PONE-D-23-37985Interventions to reduce sedentary behaviour in adults with type 2 diabetes: A systematic review and meta-analysisPLOS ONE

Dear Dr. Smith,

Thank you for submitting your manuscript to PLOS ONE. After careful consideration, we feel that it has merit but does not fully meet PLOS ONE’s publication criteria as it currently stands. Therefore, we invite you to submit a revised version of the manuscript that addresses the points raised during the review process.

**ACADEMIC EDITOR: **

We look forward to receiving your revised manuscript.

Kind regards,

Hidetaka Hamasaki

Academic Editor

PLOS ONE

2. We notice that your supplementary figures are uploaded with the file type 'Figure'. Please amend the file type to 'Supporting Information'. Please ensure that each Supporting Information file has a legend listed in the manuscript after the references list.

Reviewers' comments:

Reviewer's Responses to Questions

**Comments to the Author**

1. Is the manuscript technically sound, and do the data support the conclusions?

Reviewer #1: Partly

Reviewer #2: Yes

2. Has the statistical analysis been performed appropriately and rigorously? 

Reviewer #1: No

Reviewer #2: Yes

3. Have the authors made all data underlying the findings in their manuscript fully available?

Reviewer #1: Yes

Reviewer #2: Yes

4. Is the manuscript presented in an intelligible fashion and written in standard English?

Reviewer #1: No

Reviewer #2: Yes

5. Review Comments to the Author

Reviewer #1: Good effort was placed in this manuscript but lacks so many crucial elements to make an attractive read and source of information. It is very long with many repetitive paragraphs and ideas. The tables are huge with extensive information at makes not an optimal source of informative read. Summarizing information done for systemic review is crucial to produce a publication for readers. Many sentences also were not coherent and understood. Many subtitles of paragraphs make the article less smooth to read. I recommend a total re-wright of the manuscript.

Reviewer #2: Thank you for offering me an invitation to be the reviewer of this manuscript” Interventions to reduce sedentary behaviour in adults with type 2 diabetes: A systematic review and meta-analysis”

Siobhan et al had submitted a Systematic review and meta – analysis of literature on interventions to reduce SB in T2DM.

I am giving my observations and comments for the author to consider:

The flow of the manuscript is very comfortable. The language used is very professional and easy to follow. Text formatting may be looked into.

This systematic review and meta – analysis is observed to have followed rigorous methodology, have applied relevant search strategies and analysis methods

The types of studies included in this systematic review are thoroughly described. The PRISMA flow diagram is easy to follow and complete

Page 22 under Qualitative synthesis of non glucose outcomes: it should be “Table 2".

The discussion is detailed and elaborate.The metaanalysis findings of short term interventions and the narratives of non glucose outcomes and long term interventions are well discussed through.

All the limitations of this review are mentioned. I recommend this article may be published with the suggested minor corrections

6. PLOS authors have the option to publish the peer review history of their article (what does this mean?). If published, this will include your full peer review and any attached files.

Reviewer #1: No

Reviewer #2: No

---

## [Decision Letter · Decision Letter 1]

18 Jun 2024

Interventions to reduce sedentary behaviour in adults with type 2 diabetes: A systematic review and meta-analysis

PONE-D-23-37985R1

Dear Dr. Smith,

We’re pleased to inform you that your manuscript has been judged scientifically suitable for publication and will be formally accepted for publication once it meets all outstanding technical requirements.

Kind regards,

Hidetaka Hamasaki

Academic Editor

PLOS ONE

Additional Editor Comments (optional):

Reviewers' comments:

Reviewer's Responses to Questions

**Comments to the Author**

1. If the authors have adequately addressed your comments raised in a previous round of review and you feel that this manuscript is now acceptable for publication, you may indicate that here to bypass the “Comments to the Author” section, enter your conflict of interest statement in the “Confidential to Editor” section, and submit your "Accept" recommendation.

Reviewer #1: All comments have been addressed

2. Is the manuscript technically sound, and do the data support the conclusions?

Reviewer #1: Yes

3. Has the statistical analysis been performed appropriately and rigorously? 

Reviewer #1: Yes

4. Have the authors made all data underlying the findings in their manuscript fully available?

Reviewer #1: Yes

5. Is the manuscript presented in an intelligible fashion and written in standard English?

Reviewer #1: Yes

6. Review Comments to the Author

Reviewer #1: This is a response to the review of the paper that was submitted earlier. The authors have improved the length and flow of the information and improved significantly the table format and information.

7. PLOS authors have the option to publish the peer review history of their article (what does this mean?). If published, this will include your full peer review and any attached files.

Reviewer #1: No

---

## [Editor Report · Acceptance letter]

24 Jun 2024

PONE-D-23-37985R1 

PLOS ONE

Dear Dr. Smith, 

I'm pleased to inform you that your manuscript has been deemed suitable for publication in PLOS ONE. Congratulations! Your manuscript is now being handed over to our production team.

Kind regards, 

on behalf of

Dr. Hidetaka Hamasaki 

Academic Editor

PLOS ONE